# Myeloid Cell Leukemia 1 Small Molecule Inhibitor S63845 Synergizes with Cisplatin in Triple-Negative Breast Cancer

**DOI:** 10.3390/cancers15184481

**Published:** 2023-09-08

**Authors:** Alexus Acton, William J. Placzek

**Affiliations:** Department of Biochemistry and Molecular Genetics, University of Alabama at Birmingham (UAB), Birmingham, AL 35294, USA

**Keywords:** myeloid cell leukemia 1 (MCL1), BCL2 family, triple-negative breast cancer (TNBC), transcription factor p73, apoptosis, synergy, S63845

## Abstract

**Simple Summary:**

This manuscript builds on our recent observations that the cell-death-suppressing protein, MCL1, is able to directly regulate the DNA damage response transcription factor TP73. TP73, a homolog of TP53, is known to be activated following treatment with the anti-cancer drug, cisplatin. We therefore sought to determine if emerging anti-cancer drugs that target MCL1 combine with cisplatin in a synergistic fashion. This study provides a molecular mechanism for the prior observation that MCL1 expression can impact cisplatin sensitivity. Further, our work establishes a profile to guide further exploration of combination therapies that combine DNA damaging agents with anti-apoptotic BCL2 family inhibitors.

**Abstract:**

Triple-negative breast cancer (TNBC) is an aggressive cancer that lacks specific molecular targets that are often used for therapy. The refractory rate of TNBC to broad-spectrum chemotherapy remains high; however, the combination of newly developed treatments with the current standard of care has delivered promising anti-tumor effects. One mechanism employed by TNBC to avoid cell death is the increased expression of the anti-apoptotic protein, myeloid cell leukemia 1 (MCL1). Multiple studies have demonstrated that increased MCL1 expression enables resistance to platinum-based chemotherapy. In addition to suppressing apoptosis, we recently demonstrated that MCL1 also binds and negatively regulates the transcriptional activity of TP73. TP73 upregulation is a critical driver of cisplatin-induced DNA damage response, and ultimately, cell death. We therefore sought to determine if the coadministration of an MCL1-targeted inhibitor with cisplatin could produce a synergistic response in TNBC. This study demonstrates that the MCL1 inhibitor, S63845, combined with cisplatin synergizes by inducing apoptosis while also decreasing proliferation in a subset of TNBC cell lines. The use of combined MCL1 inhibitors with cisplatin in TNBC effectively initiates TAp73 anti-tumor effects on cell cycle arrest and apoptosis. This observation provides a molecular profile that can be exploited to identify sensitive TNBCs.

## 1. Introduction

Triple-negative breast cancer (TNBC) comprises 10–20% of all breast carcinomas and has been shown to be more aggressive with a higher reoccurrence/death rate compared to other types of breast cancer [1,2]. Part of the reason that TNBCs have limited treatment options is that they are deficient in three traditional protein targets of breast cancer therapy—the estrogen receptor (ER); the progesterone receptor (PR); and the human epidermal growth factor receptor 2 (HER2) [3,4,5]. The main systemic treatment options for TNBC are combinations of neoadjuvant chemotherapies including taxanes, alkylating agents, and platinum-based chemotherapeutics. Yet, TNBC quickly develops resistance to these treatment strategies [6,7,8]. To overcome resistance and improve therapy, recent clinical trials investigated a combination treatment of immune-activating PD-L therapies with platinum-based or taxane chemotherapies [9,10,11]. The success of these trials highlights that the combination of targeted therapies with the standard of care can advance the treatment of TNBC.

Multiple mechanisms can contribute to a cancer cell’s ability to resist cell death and thereby enable tumorigenesis. One such mechanism is the upregulation of anti-apoptotic Bcl-2 family proteins in response to cellular stress, such as that induced by neoadjuvant chemotherapy. Emerging BH3 mimetics that specifically target the hydrophobic binding groove of the anti-apoptotic Bcl-2 proteins directly target this resistance mechanism [12,13,14]. Small molecule Bcl-2 family inhibitors act as antagonists of anti-apoptotic proteins to sensitize cells to intrinsic stress. ABT-199 (Venetoclax), the first BH3 mimetic approved to treat chronic lymphocytic leukemia (CLL) in 2016, specifically targets BCL2 [15,16,17]. However, molecules that were initially designed to target BCL2 have shown acquired resistance due to the upregulation of MCL1 [18,19]. In addition, the upregulation of the BCL2 homolog MCL1 is more common in TNBC and has been shown to be correlated with an increased tumor size and invasion [20,21,22,23]. Partly, as a result, BH3 mimetics specifically targeting MCL1 have begun clinical development. A leading MCL1 small molecule inhibitor in phase II clinical trials, S63845, binds with a high affinity and specificity to MCL1′s BH3 binding groove over other anti-apoptotic family members [24,25]. As the upregulation of MCL1 is common in TNBC, we sought to determine if these emerging inhibitors can be combined with neoadjuvant chemotherapies to improve the anti-cancer response.

Multiple classes of broad-spectrum anti-cancer agents can be used in the treatment of TNBC. Platinum-based compounds have been shown to be dependent on MCL1 for both response and resistance [26,27]. These chemotherapies, e.g., cisplatin or carboplatin, are DNA chelators that deliver platinum to cells where it intercalates into DNA and thereby induces double-stranded DNA breaks (DSB) [28]. These DSB induce the downstream activation of the tumor suppressor family TP53 to drive the response element activation of the cell cycle check point, cellular apoptosis, autophagy, and senescence [29,30]. Prior studies have identified that a significant driving factor in the majority of sporadic breast cancers is the mutation or loss of TP53 [31,32]. Thus, the platinum response in TNBC often relies on the TP53 family homolog TP73 for anti-tumor effects. The TP73 gene results in the expression of numerous N- and C-terminal isoforms; in this study, we focus on the pro-apoptotic TAp73 versions of TP73. Recently, we demonstrated that the anti-apoptotic protein, MCL1, binds and negatively regulates TP73 (both TAp73 and ΔNp73) through a unique rBH3 motif located in its tetramerization domain (TD) [33]. This leads to the effect in which the upregulation of MCL1 in these cancers not only suppresses the traditional Bcl-2 family regulation of apoptosis, but also suppresses the specific upregulation of TP73 and the resulting anti-tumor effects of these platinum-based compounds. We therefore sought to see if the inhibition of MCL1 combined with cisplatin treatment could enhance the activation of TP73, thereby inducing pro-death signaling.

Here, we build on recent studies that define MCL1’s ability to bind TP73 and demonstrate MCL1’s ability to thereby impact the cell cycle progression and apoptosis in TNBC [33]. We hypothesized that as both TP73 and MCL1 are essential cell fate regulators in TNBC, targeting this signaling axis could provide a new treatment strategy for TNBC. We observe that S63845 and cisplatin synergize in TNBC cell lines. Further, we expand on MCL1’s ability to suppress TP73 and demonstrate that the inhibition or knockdown of MCL1 enhances the TP73 downstream transcriptional activity. This suggests that the combination of MCL1 inhibitors with platinum-based chemotherapies could be beneficial for TNBC.

## 2. Materials and Methods

### 2.1. Cell Culture

MDA-MB-468, HCC1143, MDA-MB-231, MDA-MB-436, and Hs-578T cells were incubated in a humidified atmosphere of 5% CO_2_ at 37 °C. Cells were supplemented with RPMI 1640 medium (Invitrogen, Waltham, MA, USA), 1× antibiotic–antimycotic (Invitrogen, Waltham, MA, USA), and 10% fetal bovine serum (FBS) (ThermoScientific, Waltham, MA, USA). All cell lines were purchased from ATCC and validated at the Heflin Center Genomics Core Facility at UAB via STR profiling. All cells were passaged on a rotation of a 4:3-day cycle. Cells were rinsed with 5 mL of sterile Dulbecco’s phosphate-buffered saline (1× dPBS) (ThermoScientific, Waltham, MA, USA) prior to trypsin dissociation or lysis. All cell lysates were prepared in 1× RIPA lysis buffer (Pierce, Waltham, MA, USA, 87788) supplemented with 1× Halt Protease Inhibitor Cocktail with EDTA (ThermoScientific, Waltham, MA, USA, 1861279).

### 2.2. Proliferation and Viability Assays

#### 2.2.1. Growth Curves and Trypan Blue Viability Staining

MDA-MB-468 (sensitive) and MDA-MB-231 (resistant) cells were seeded at a density of 2.5 × 10^5^ cells in a 6-well plate in RPMI 1640 (1×) supplemented with 2.5% FBS to analyze the impact of cisplatin (Selleck Chemicals, Houston, TX, USA) and S63845 (Chemietek, Indianapolis, IN, USA) on growth and viability. Cells were placed into a humidified atmosphere of 5% CO_2_ at 37 °C for 24 h before the medium was replaced and cells were dosed with various chemotoxic treatments. Technical triplicate cell counts were taken every 24 h throughout the 96 h timepoint for each treatment condition. Cells were collected through tryspin dissociation and washed with 1× dPBS. Equal parts of cell suspension and Trypan blue were mixed, and cell numbers and percent viability were recorded on a BioRad Automated Cell Counter (BioRad Laboratories, Hercules, CA, USA).

#### 2.2.2. Colony Formation Assay

MDA-MB-468 and MDA-MB-231 cells were seeded in a 6-well plate in RPMI 1640 (1×) supplemented with 2.5% FBS at a density of 1 × 10^4^ cells/well. Twenty-four hours post treatment, cells were treated with various concentrations of cisplatin and S68345 and incubated in a humidified atmosphere of 5% CO_2_ at 37 °C for 6 days. Medium was removed, and cells were washed with 1× dPBS fixed with 4% formaldehyde at 4 °C for 30 min while rocking. Cells were washed with 1× dPBS and incubated in 0.5% crystal violet at room temperature for 1 h. Staining solution was removed, and cells were washed 4 times with 1× dPBS. Cells were imaged on an Olympus IX51 Inverted Fluorescence microscope (Olympus, Tokyo, Japan). Images were quantified using Fiji Image J (imagej.net; Version 1.0) where colonies >5 cells were considered for analysis.

#### 2.2.3. MTS Assay Kit (Cell Proliferation Colorimetric Assay)

Cells were seeded in a 96-well plate in RPMI 1640 (1×) (- Phenol Red) supplemented with 2.5% FBS. Cells were placed into a humidified atmosphere of 5% CO_2_ at 37 °C for 8 h and then dosed on a 9-point dose–response curve with a final DMSO/DMF concentration of 0.25%. After cells underwent incubation for 72 h, 10 μL of MTS assay reagent (ab197010) was added and incubated at 37 °C for 1 h. The 96-well plate was read on a PerkinElmer Victor X5 Multimode Plate Reader at 490 nm.

### 2.3. Western Blot Analysis

Cells were seeded at a density of 2.0 × 10^5^ cells/mL for 24 h prior to chemotoxic treatment. Cells were lysed as explained above. An amount of 4× Laemmli sample buffer containing β-mercaptoethanol (ThermoScientific, Waltham, MA, USA) was added to each sample and denatured at 95 °C for 10 min. An SDS polyacrylamide gel electrophoresis at 150 V for 45 min was utilized to resolve all proteins and transferred on a PVDF membrane using a TransBlot Turbo semi-dry transfer system (BioRad Laboratories, Hercules, CA, USA) for 7 min at 1.3 A and 25 V. Five percent milk *w*/*v* nonfat milk in phosphate-buffered saline + tween (PBST) was used as a blocking agent for 1 h post transfer. Primary antibodies were incubated overnight at 4 °C. Blots were then washed with 1× PBS supplemented with 0.01% tween. All secondary antibodies were incubated at room temperature for 1 h, and PBST washes were repeated. All Western blots were exposed via ECL reagent (Pierce, Waltham, MA, USA). A BioRad ChemiDoc MP imaging system was utilized to image all Western blots. The original western blot figures could be found in Appendix A.

### 2.4. Antibodies

#### 2.4.1. Primary Antibodies

All antibody dilutions for protein detection were as follows: MCL1 protein: anti-MCL1 rabbit mAb (D2W9E, Cell Signaling, Danvers, MA, USA); TAp73 protein: anti-TAp73 mouse mAb (5B429, Novus Biologicals, Centennial, CO, USA), diluted to 1:500; GAPDH protein: anti-GAPDH XP rabbit mAb (D16H11, Cell Signaling, Danvers, MA, USA); cleaved caspase 3 (CC3) protein: anti-CC3 rabbit mAb (9664S, Cell Signaling, Danvers, MA, USA). All antibodies were diluted in 1% milk-PBST to a ratio of 1:1000 unless stated otherwise.

#### 2.4.2. Secondary Antibodies

All secondary antibodies were diluted to 1:2000. Rabbit: goat anti-rabbit IgG-HRP (Cell Signaling, Danvers, MA, USA). Mouse: horse anti-mouse IgG-HRP Cell Signaling, Danvers, MA, USA).

### 2.5. RNA Extraction and cDNA Synthesis

RNA was purified for RT-qPCR with the E.Z.N.A Total RNA Kit I (Omega Bio-Tek, Norcross, GA, USA, R6834-01) per the manufacturer’s protocol. The final concentration was measured using a Nanodrop 2000c spectrophotometer (ThermoScientific, Waltham, MA, USA). A cDNA synthesis reaction was performed per the qScript cDNA SuperMix (Quanta Biolabs, Beverly, MA, USA) manufacturer’s protocol. Final concentration of the cDNA was determined on a Nanodrop 2000c spectrophotometer. All cDNA was diluted to 25 ng/μL with nuclease-free water 30 min prior to RT-qPCR analysis. Remaining RNA was stored at −80 °C.

### 2.6. TaqMan RT-qPCR

All RT-qPCR reactions were performed on a CFX Opus 384 Real-Time PCR system (BioRad Laboratories, Hercules, CA, USA). A master mix for each target was made composing of 20× RT-qPCR Primer Mix (ThermoScientific, Waltham, MA, USA), cDNA as described above, and TaqMan Universal Master Mix II with UNG (Applied Biosystems, Waltham, MA, USA). All reaction wells (100 ng/reaction) were performed in technical triplicate. Each cycle was as follows: 50 °C for 2 min, 95 °C for 10 min and 15 s, and 60 °C for 1 min. Each step was repeated 39× for a total of 40 cycles. All target gene Cp values were normalized to their respective treatment’s housekeeping gene, GAPDH. Each treatment group was then normalized to the DMSO/DMF control as a relative fold change of 1. Error propagation was calculated through a ddCp value for each gene target. All primers had FAM-MGB probes, ordered from Thermo Fisher Scientific (Waltham, MA, USA) as follows: (Hs00355782_m1):CDKN2A/p21, (Hs00560401_m1):PMAIP1/NOXA, (Hs00169255_,1):GADD45A, and (Hs0275899_g1):GAPDH. All RT-qPCR reactions were performed in biological triplicate.

### 2.7. Annexin V/PI Staining with Fluorescence-Activated Cell Sorting (FACS)

Cells were seeded 1 × 10^6^ in a 10 cm dish and treated through various conditions to modulate TAp73 and MCL1 expression. After 24 h of drug treatment, medium was removed and placed in a labeled 15 mL falcon tube. Tryspin dissociation was used to collect cells, and cells were washed with 1× dPBS and placed in corresponding tube. All cell counts were performed on a BioRad Automated Cell Counter (BioRad Laboratories, Hercules, CA, USA). Cells were pelleted for 5 min at 1000× *g*. Each FACs tube contained 5 μL propidium iodide (PI) staining solution (BD Pharmigen, San Diego, CA, USA, 51-66211E) and 5 μL FITC-Annexin V (BD Pharmigen, San Diego, CA, USA, 556419). A total of 5 × 10^5^ cells were resuspended in 1× Annexin V binding buffer (BD Pharmingen, 556454) and transferred to the corresponding FACs. Cells were incubated at RT in the dark for 15 min. After incubation, 450 μL of 1× Annexin V Binding buffer was added, and FACS was collected on a BD LSRFortessa (BD Biosciences, Franklin Lakes, NJ, USA) and analyzed using FlowJo (flowjo.com; version 10). Compensation controls used for this analysis included unstained cells, and cells stained for each individual fluorophore (Cy5-PI and FITC-Annexin V). Gating strategy eliminated cell debris and doublets through forward and side scatter plots.

### 2.8. Propidium Iodine (PI) Staining

Cells were seeded in a 6-well plate and treated through various conditions to modulate TAp73 and MCL1 expression. After 24 h of drug treatment, medium was removed and placed in a labeled 15 mL falcon tube. Cells were collected through trypsin dissociation, washed with 1× dPBS, and placed in corresponding tube. Cells were pelleted at 500× *g* 5 min and resuspended in 1× dPBS for cell counts, and then re-pelleted. Cell pellets were resuspended in 500 μL in 1× dPBS + 1% formaldehyde. Cells were vortexed as 4.5 mL of 70% EtOH was added dropwise. Lysate was stored at −20 °C for 2 h. Cells were washed with 5 mL 1× dPBS and pelleted at 1500× *g* 10 min. Cells were resuspended in 200 μL 1× dPBS + 100 μg/mL RNase A and incubated in a 37 °C water bath for 30 min. A total of 100 μL RNased sample was transferred to FACS tube with 200 μL or 50 μg/mL of PI stain. BD LSRFortessa was used to collect all FACS and they were analyzed on FlowJo v10. Unstained and single-stained controls were used for all compensation controls for analysis. The elimination of all doublets and cell debris was excluded in the forward and side scatter gating strategies. All data were collected on separate days with technical duplicates. Each analysis was performed in biological triplicate.

### 2.9. siRNA Transfection

Cells were seeded at 2.5 × 10^5^ cells/well in a 6-well plate 24 h post siRNA knockdown. All RNA interference transfections were performed using Lipfectamine RNAiMax (Invitrogen, Waltham, MA, USA) per the Lipfectamine RNAiMAX manufacturer’s protocol. Final concentrations of siRNA-targeting MCL1 and TAp73 were 10 pmol/well and 100 pmol, respectively. siRNA (Ambion, Austin, TX, USA) silencer selected the following sequences for MCL1: siMCL1 (s8585)—GTAATTAGGAACCTGTTTCtt and siMCL1 #2 (s8583)—CCAGUAUACUUCUUAGAAAtt. siRNA (Ambion, Austin, TX, USA) silencer selected the following sequence for TAp73: siTAp73 (s14320)—CCACCAUCCUGUACAACUUtt.

#### 2.9.1. Reagents

S63845 (Chemietek, Indianapolis, IN, USA, S-63845): MCL1-specific inhibitor, Venetoclax (ABT199) (Selleckchem, Houston, TX, USA S-8048), and cisplatin (SelleckChem, Houston, TX, USA S1166). All other reagents are described above.

#### 2.9.2. Statistical Analyses

All experiments comprise biological triplicates with two to three technical replicates as reported. No samples were excluded, and all data are expressed as the mean ± SD. Statistical significance was performed in Prism (GraphPad Inc., San Diego, CA, USA); * *p* < 0.05, ** *p* < 0.01, and *** *p* < 0.001.

## 3. Results

### 3.1. Cisplatin and S63845 Synergize in TNBC

As a first step to investigating if the MCL1-TP73 axis impacts the cisplatin sensitivity in TNBC, we determined the sensitivity of each of our TNBC cell lines to single-agent cisplatin or S63845 treatment. For our studies, we chose three well-characterized TNBC cell lines classified as having alterations in TP53 (MDA-MB-468, MDA-MB-231, and Hs578t) and one cell line with WT TP53 (MDA-MB-436). All chosen cell lines exhibited an amplification of MCL1 [20]. Two basal-like TNBC cell lines, MDA-MB-468 and HCC1143, exhibited sensitivity to both cisplatin (EC_50_ 492.1 ± 43.5 nM and 16.76 ± 0.05 μM, respectively) and S63845 (EC_50_ 141.2 ± 24.7 nM and 3.1 ± 0.5 μM, respectively) (Figure 1A,B,D,E). The mesenchymal cell lines, MDA-MB-231, Hs-578T, and MDA-MB-436, were resistant to both single-agent treatments (Figure 1C,F and Appendix A). For these studies, resistance was classified as an EC_50_ > 20 μM for cisplatin and an EC_50_ > 4 μM for S63845.

As cisplatin resistance and toxicity decreases its efficacy in a clinical setting, we sought to determine if cisplatin combined with an MCL1-targeted inhibitor is synergistic. For these studies, we tested three concentrations of S63845 guided by the single agent analysis. MDA-MB-468 cells co-treated with cisplatin and 30, 60, or 100 nM of S63845 exhibited a decrease in the effective EC_50_ value (Figure 2A–C).

A similar synergistic effect was observed in HCC-1143 cells (Appendix A). Isobologram visualization between S63845 and cisplatin in both MDA-MB-468 and HCC-1143 cell lines demonstrate synergy (Figure 2D and Appendix A). Since MDA-MB-231, Hs-578T, and MDA-MB-436 showed resistance to cisplatin, we hypothesized that an MCL1 inhibitor may re-sensitize these lines to cisplatin. Regrettably, none of these cell lines exhibited any change in sensitivity to cisplatin when co-treated with 500 nM S63845 (Appendix A).

### 3.2. Synergy Is Dependent on MCL1 Inhibition

Prior studies have shown that the MCL1 binding pocket is highly selective to rBH3 interactions, making the MCL1 BH3 binding groove unique compared to other anti-apoptotic family members. While we know that S63845 has been well characterized to specifically bind to MCL1, we wanted to analyze the inhibition of another anti-apoptotic member to determine if synergy is unique to MCL1 inhibition or if it is a hallmark of any anti-apoptotic Bcl-2 inhibitors. To achieve this, we employed the clinically approved BCL2 inhibitor ABT-199. Single-agent treatment in MDA-MB-468 and HCC-1143 resulted in EC_50_ values of 2.6 ± 0.952 μM and 3.0 ± 0.157 μM, respectively (Figure 2E and Appendix A). We then analyzed ABT-199 in combination with cisplatin, where we determined that there was no dose-dependent decrease in the EC_50_ values of cisplatin (Figure 2F and Appendix A). Thus, cisplatin and ABT-199 exhibited no synergistic effects when co-treated in either of the TNBC cell lines. Together, these results suggest that MDA-MB-468 and HCC-1143 are uniquely sensitive to MCL1 inhibition in combination with cisplatin.

### 3.3. Cisplatin and S63845 Decreases Proliferation in MDA-MB-468 Cells

The molecular mechanisms of cisplatin and S63845 target two essential proteins involved in cell fate decisions. Our prior studies identified that MCL1 binds TP73 and inhibits its association with DNA [33]. As TAp73 is a transcription factor that regulates the cell cycle checkpoint proteins p21 and GADD45, we hypothesized that treatment with cisplatin would result in decreased proliferation. We also sought to investigate how the inhibition of MCL1 would impact proliferation.

Utilizing the EC_50_ values calculated from the initial screenings, we assessed the viable cell numbers over 96 h using trypan blue staining and cell counting. We observed a significant decrease in the viable cell numbers in MDA-MB-468 cells following single-agent treatment with 400 nM cisplatin or 100 nM S63845 (Figure 3A). We next sought to determine if lower doses of either compound identified in our isobologram analysis impact proliferation as either a single agent and/or in combination. We observed a significant decrease in viable cells following treatment with 30 nM S63845 at 96 h (Figure 3B,C). However, more significant decreases were observed following treatment with 100 nM cisplatin alone or in combination with 30 nM S63845 (Figure 3B,C). In comparison, the resistant cell line MDA-MB-231 exhibited a continuous increase in viable cell numbers over 72 h in both the cisplatin- and S63845-treated groups (Figure 3D and Appendix A). Together, these data indicate that cisplatin in combination with S63845 has a greater anti-proliferative effect in MDA-MB-468 cells compared to single-agent treatments of cisplatin or S63845.

Due to the decrease in the viable cell number observed in the MDA-MB-468 cells after various treatments, we sought to determine if there was a change in the cell cycle distribution. In order to look at the cell cycle stages, we utilized propidium iodine (PI) cell cycle staining to look at the populations of cells in G1, S, G2/M, or sub G1 24 h post treatment. The MDA-MB-468 cells treated with cisplatin exhibited an increase in the G2/M population, indicative of an induction of a G2/M cell cycle arrest that was previously identified as a result of cisplatin treatment (Figure 3E). Conversely, treatment with S63845 significantly increased the sub-G1 population, suggesting that S63845 impacts viability and not proliferation (Figure 3E). A combination of the two drugs amplified both of these results with increases in the S-phase cells and sub-G1 cells (Figure 3E). In line with ABT-199 exhibiting a minimal dose-dependent impact on the cell viability (Figure 2E,F), ABT-199 had no impact on the cell cycle stages in the MDA-MB-468 cells (Figure 3E). Similar results post cisplatin or combination treatment were observed in the G2/M cell population in MDA-MB-231 cells (Figure 3F).

### 3.4. Cisplatin and S63845 Induce Apoptosis in MDA-MB-468 and HCC1143 TNBC Cell Lines

The PI cell cycle analysis demonstrated that treatment with cisplatin and/or S63845 induced a variable increase in the proportion of sub-G1 cells, indicative of the changes in the cellular apoptosis induction. Since MCL1 is a critical anti-apoptotic protein involved in regulating intrinsic apoptosis, we sought to investigate the amount of apoptotic cell death post treatment using Annexin V/PI flow cytometry.

For these studies, we treated MDA-MB-468 with 100 nM cisplatin or 30 nM S63845 for 24 h. All treatments induced a significant increase in the cell population in the early apoptotic quadrant, indicated by Q3 in Figure 4. As a single agent, the cisplatin and S63845 treatment groups exhibited 4.1% and 21.8% apoptotic cells, respectively. In line with our prior synergy analysis, a combination of cisplatin and S63845 synergistically induced a higher portion of early apoptotic cells (33.8%) in the Q3 quadrant compared to the single-agent experimental groups. Similar results were observed in our other sensitive cell line HCC-1143 (Figure 4B,E). In contrast, the MDA-MB-231 cells treated with either cisplatin or S63845 showed no change in the early apoptotic cell population (Appendix A).

To confirm the cell death induction of S63845 and the combination treatment, we assessed the cellular levels of cleaved caspase 3 (CC3) after both single-agent and combination treatments. The treatments with S63845 alone and in combination with cisplatin induce the expression of CC3 in both cell lines (Figure 3D,F), thus supporting the hypothesis that S63845 and cisplatin synergistically induce apoptosis in two TNBC cell lines.

### 3.5. Combination of Cisplatin and S63845 Prevents Colony Formation in MDA-MB-468 Cells 6 Days Post Treatment

Due to the significant increase in the sub G1 population observed in our PI cell cycle staining and the increase in the early apoptotic cell population in the MDA-MB-468 cells, we wanted to further analyze the impacts that the single-agent treatment or co-treatment of cisplatin and S63845 have on long-term cell viability.

To achieve this, we employed a colony formation assay and assessed colony growth 6 days post treatment. Notably, we observed that MDA-MB-468 had a significant decrease in colonies in the cisplatin and combination groups but only a slight reduction in colonies following treatment with S63845 alone (Figure 5A. In line with other proliferative assays, the MDA-MB-231 cells did not show a significant change in the ability to form colonies post treatment (Figure 5B). Together, these data suggest that while S63845 promotes a pro-death phenotype, cisplatin treatment significantly impacts cell replication after initial cell death. Additionally, our combination studies highlight that S63845 not only induces an initial induction of apoptosis, but it also improves cisplatin’s long-term anti-proliferative effects in MDA-MB-468 cells.

### 3.6. TAp73 Mediates Cell Cycle Halt and Apoptotic Gene Expression Post Treatment with Cisplatin and S63845

The decrease in the viable cell counts in our growth curve analysis, as well as the increase in the apoptotic cell population, led us to investigate the known gene targets involved in regulating these pathways. This led us to our hypothesis that treatment with cisplatin, S63845, or a combination of both would induce the expression of the TAp73 protein as well as the TAp73 target genes involved in cell cycle checkpoints and apoptosis. In order to test this, we utilized RT-qPCR post treatment in MDA-MB-468 and MDA-MB-231 cell lines.

MDA-MB-468 cells treated over varying concentrations of cisplatin and S63845 exhibited significant increases in TAp73, p21, NOXA, and GADD45 expressions (Figure 6A). In contrast, the resistant cell line MDA-MB-231 did not exhibit a significant change in gene expression post treatment (Figure 6B). The increase in the TAp73 gene expression then led us to investigate the protein levels in both cell lines. The MDA-MB-468 cells treated with 400 nM cisplatin or 100nM cisplatin combined with 30 nM S63845 exhibited a significant increase in the TAp73 expression (Figure 6C). In MDA-MB-231, we observed an increase in the TAp73 expression in both treatment groups (Figure 6D). Both cell lines showed a stabilization of MCL1 upon treatment with S63845, which was well characterized in previous studies (Figure 6C,D) [24,34].

### 3.7. TAp73 and MCL1 Mediate Synergistic Effects with Cisplatin and S63845

The expression of TAp73 and its targets significantly increase post treatment with S63845. However, general anti-apoptotic protein inhibition could cause robust changes in gene expression irrespective of the MCL1-p73 axis. Therefore, we wanted to validate that this activation of TAp73 was due to MCL1 inhibition and not off-target cytotoxicity. To achieve this, we knocked down MCL1 with two siRNAs and analyzed the impact on TAp73, p21, NOXA, and GADD45. The MDA-MB-468 cells treated with siMCL1 showed an increased gene expression of TAp73 and its downstream targets (Figure 7A,B and Appendix A). Likewise, the cells treated with siMCL1 and 100 nM of cisplatin exhibited a more significant increase in all targets compared to our negative control siGFP treated with 100 nM of cisplatin (Figure 7C and Appendix A). Together, these data suggest that that in the absence of MCL1, TAp73 has a more robust activation and response element regulation after cisplatin treatment.

We next sought to validate whether the gene activation following MCL1 inhibition with S63845 was TAp73-dependent. In order to achieve this, we knocked down TAp73 with an siRNA. We observed a loss of both TAp73 protein and mRNA using Western blot and RT-qPCR analyses, respectively (Figure 7D,E). We then analyzed the gene expression induced by S63845 treatment after either siGFP or siTAp73 knockdown. We observed that treatment with S63845 did not lead to the activation of TAp73, p21, NOXA, or GADD45 in the siTAp73-treated cells (Figure 7F), further validating that gene activation is solely dependent on TAp73. Since cisplatin is a known TAp73 activator, we next sought to determine how treatment with cisplatin affects proliferation and viability in MDA-MB-468 cells transfected with siTAp73. To achieve this, we utilized growth curve and trypan blue staining assays over 72 h in MDA-MB-468 cells. We observed a significant increase in the viable cell numbers and trypan blue percent viability in our siTAp73 + cisplatin compared to our off-target siGFP + cisplatin control, further validating that the decreased proliferation and viability response to cisplatin observed in Figure 3A–C was mediated by TAp73 (Appendix A).

## 4. Discussion

In this study, we described how combination therapy between an MCL1 inhibitor, S63845, and cisplatin synergize in a subset of TNBC cell lines. Further, this study identified the molecular mechanism by which these drugs synergize to promote cell cycle arrest and apoptosis. Multiple studies have outlined how increased MCL1 regulates cisplatin sensitivity. These strategies include the suppression of CHK1, the suppression of TP73, and its canonical anti-apoptotic function [33,35,36,37]. This suggests that co-treatment with an MCL1 inhibitor and cisplatin could provide improved therapy compared to cisplatin treatment alone. Ultimately, we observed that S63845 synergizes with cisplatin in basal-like TNBC lines and specifically induces a pro-apoptotic state while amplifying the anti-proliferative activity of cisplatin. Further, we demonstrated that the upregulation of the TP73 target gene expression induced by S63845 treatment is TP73-specific.

It was reported that the activation of TP73 is crucial for the transactivation of response elements that regulate the DNA damage response [38,39]. While platinum-based chemotherapies have been characterized to activate the TP73 family of transcription factors, the developed resistance to platinum remains a huge barrier in TNBC. Recent advancements were made in understanding how MCL1 contributes to dsDNA break (DSB) repair both via colocalization with essential repair machinery (e.g., 53BP1) as well as the negative regulation of the transactivation of TP73 targets involved in DSB repair, apoptosis, and cell cycle progression [27,33,40,41,42]. MCL1′s ability to prevent TP73 activation could mechanistically explain prior studies investigating how MCL1 expression is a key determinant in platinum-based chemotherapeutic response. Therefore, it is not surprising that the inhibition of MCL1 with S63845 improves the cisplatin response in some TNBC cell lines.

We found a distinct difference in the response to cisplatin and S63845 between cell lines classified as either basal-like or mesenchymal-like TNBC according to the ATCC. Prior studies have shown that basal-like subtypes have up to four times higher clinical remission rates than mesenchymal-like TNBC in response to taxane-based chemotherapies [43,44]. Mesenchymal cell lines (MDA-MB-231, Hs-578T, and MDA-MB-436) express high levels of MCL1 but have no response to MCL1 inhibition both in monotherapy and in combination therapy with cisplatin [20,45]. Mesenchymal-like TNBC may be treated with drugs targeting the epithelial–mesenchymal transition (EMT). A study performed in 2018 by Inao et al. combined doxorubicin, an epithelial–mesenchymal transition inhibitor, and ABT-199 and observed decreases in proliferation and increases in autophagy and apoptosis [46]. EMT enhances drug resistance due to increased drug efflux and alterations in signaling pathways involved in proliferation and apoptosis, and thus, combination therapy with other Bcl-2 family inhibitors and anthracyclines need further investigation [47,48,49].

The evolution of novel targeted therapeutic agents remains a priority for TNBC. With the refinement of TNBC subtypes over the past 10 years, the use of old drugs in combination with newly developed chemotherapies is an important area of research that can improve clinical outcomes. In this study, we identified both TP73 and MCL1 as molecular biomarkers that can be targeted with specific chemotherapies. Additionally, while this study focuses on TNBC, the mechanisms discovered may apply to other cancer types. Pancreatic and ovarian cancers have both been shown to have amplified levels of MCL1, with one of the mainstay therapies being platinum-based reagents [20,50,51,52,53]. Thus, the combination of MCL1 inhibitors with platinum reagents could enhance the clinical response rate.

## 5. Conclusions

In conclusion, we demonstrate a synergistic effect of the MCL1 inhibitor S63845 in combination with cisplatin in TNBC cell lines. Given that platinum-based resistance in patients with TNBC has been a therapeutic hurdle, our findings suggest that the combination with MCL1 inhibitors could overcome this barrier to improve the platinum efficacy in TNBC patients. Our data suggest that lower doses of cisplatin and S63845 show enhanced anti-tumor effects compared to that of high-dose cisplatin administration. Finally, our results suggest that determining the expression of MCL1 and TP73 may identify a subset of TNBC patients that would benefit from the combination treatment of platinum and emerging MCL1 inhibitors.

## Figures and Tables

**Figure 1 cancers-15-04481-f001:**
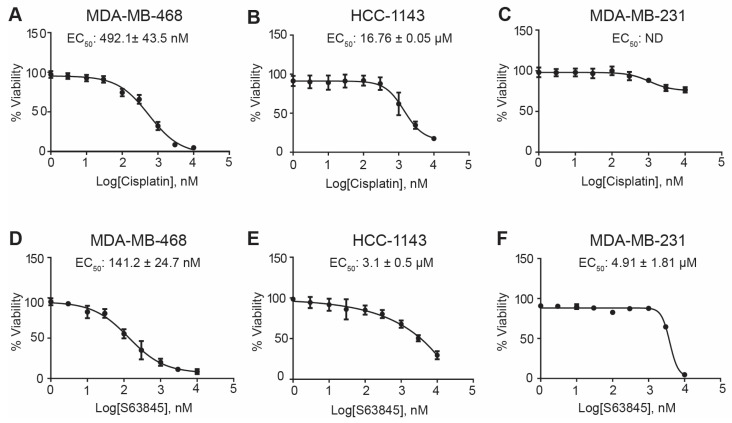
MDA-MB-468 and HCC1143 TNBC cell lines show response to cisplatin and S63845. (**A**) MDA-MB-468, (**B**) HCC-1143, and (**C**) MDA-MB-231 cells were treated on a 9-point dose–response curve with cisplatin and (**D**–**F**) S63845 for 72 h, respectively. Percent viability was analyzed with an MTS assay. EC_50_ values were interpolated using a nonlinear regression on Prism GraphPad. All experiments were performed in biological triplicate.

**Figure 2 cancers-15-04481-f002:**
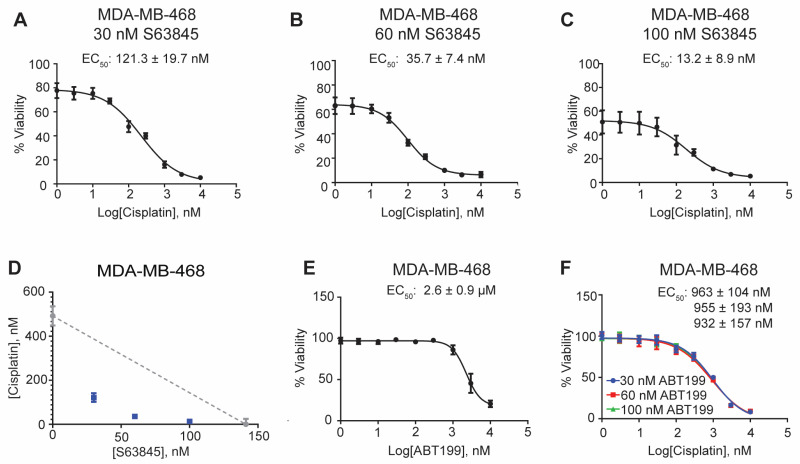
Cisplatin and S63845 synergize in MDA-MB-468 cells. MDA-MB-468 cells were treated with (**A**) 30 nM, (**B**) 60 nM, and (**C**) 100 nM S63845 over a 9-point dose–response of cisplatin for 72 h. Isobologram analysis of (**D**) MDA-MB-468 cells representing both single-agent (grey) and combination (blue) EC_50_ values. (**E**) MDA-MB-468 cells were treated with ABT-199 for 72 h. (**F**) MDA-MB-468 combinational studies of various ABT-199 concentrations over a 9-point dose–response of cisplatin. Percent viability was analyzed through MTS assay, and EC_50_ values were interpolated using a nonlinear regression on Prism GraphPad. All experiments were performed in biological triplicate.

**Figure 3 cancers-15-04481-f003:**
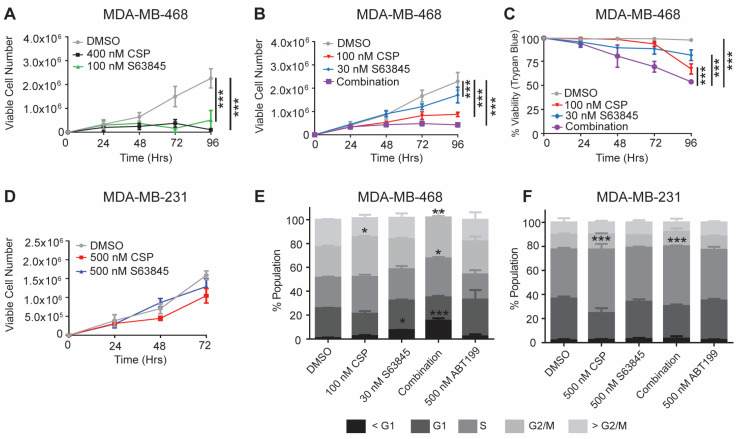
Cisplatin and S63845 decrease cellular proliferation in MDA-MB-468 cells. Growth curve analysis of MDA-MB-468 cells treated with both (**A**) single agent and (**B**) combined EC_50_ concentrations of cisplatin and S63845 over 96 h. (**C**) Matched sample trypan blue staining of MDA-MB-468 cells treated with various concentrations of cisplatin and S63845. (**D**) Growth curve analysis of MDA-MB-231 cells treated with either 500 nM cisplatin or S63845 over 72 h (confluent). Propidium iodine (PI) staining of (**E**) MDA-MB-468 and (**F**) MDA-MB-231 cells treated over various conditions for 24 h. Differences between groups were evaluated using a matched two-way ANOVA followed by Tukey’s post hoc test; *p* < 0.05. * *p* < 0.05; ** *p* < 0.01; and *** *p* < 0.001. All experiments were performed in biological triplicate.

**Figure 4 cancers-15-04481-f004:**
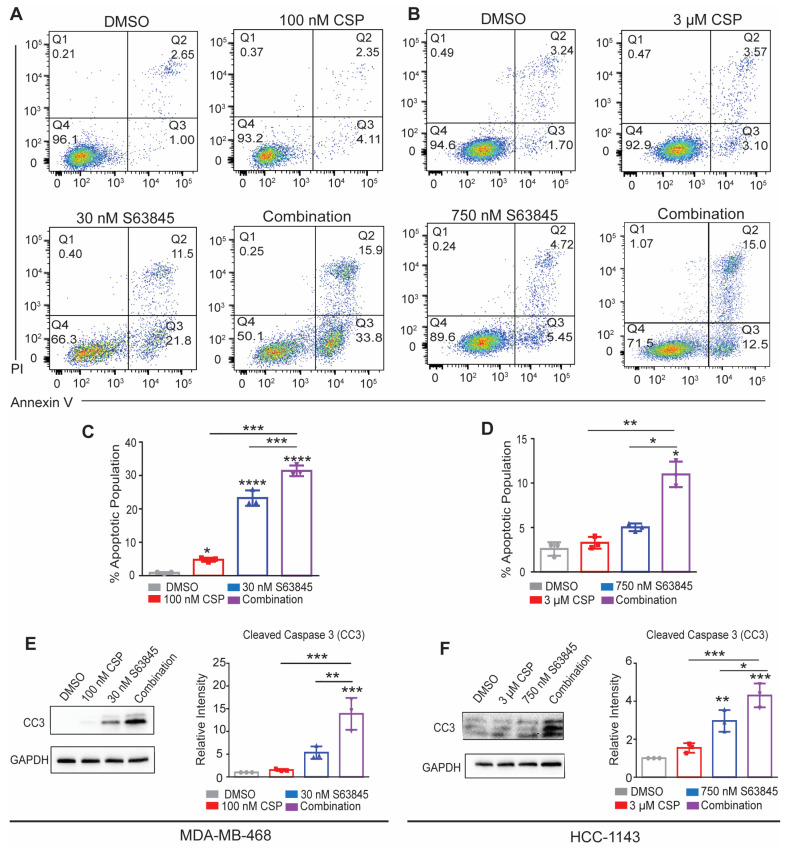
Combination therapy increases early apoptotic cell population in MDA-MB-468 and HCC-1143 cells. Annexin V/PI staining of (**A**) MDA-MB-468 and (**B**) HCC-1143 cells treated with combined EC_50_ concentrations of cisplatin and S63845 for 24 h. (**C**) MDA-MB-468 and (**D**) HCC-1143 cells’ Annexin V/PI Q3 quantification of early apoptotic cell population percentage. Western blot analysis of (**E**) MDA-MB-468 and (**F**) HCC-1143 cells treated over various conditions for 24 h. Differences between groups were evaluated using a matched two-way ANOVA followed by Tukey’s post hoc test; *p* < 0.05. * *p* < 0.05; ** *p* < 0.01; *** *p* < 0.001; **** *p* < 0.0001. All experiments were performed in biological triplicate.

**Figure 5 cancers-15-04481-f005:**
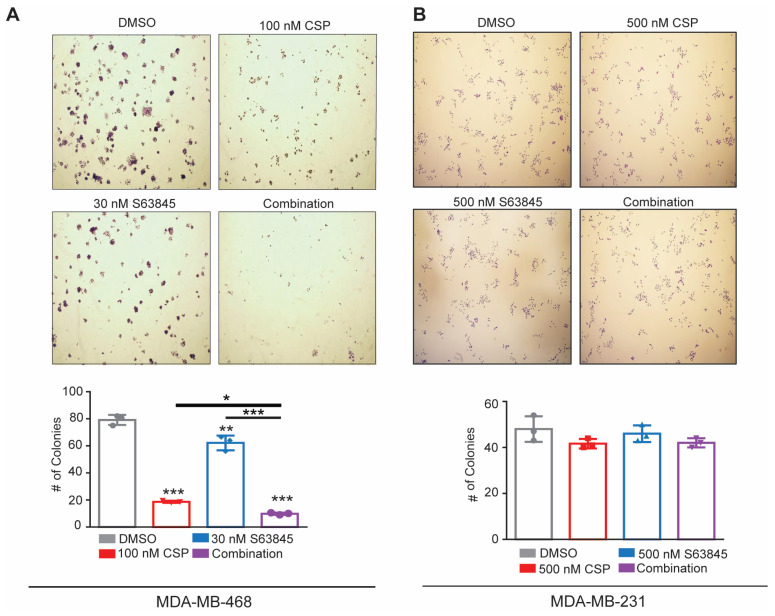
Cisplatin alone or in combination with S63845 decreases colony formation in MDA-MB-468 cells. Colony formation assays of (**A**) MDA-MB-468 and (**B**) MDA-MB-231 cells treated over various conditions for 6 days. Differences between groups were evaluated using a matched two-way ANOVA followed by Tukey’s post hoc test; *p* < 0.05. * *p* < 0.05; ** *p* < 0.01; *** *p* < 0.001. All experiments were performed in biological triplicate.

**Figure 6 cancers-15-04481-f006:**
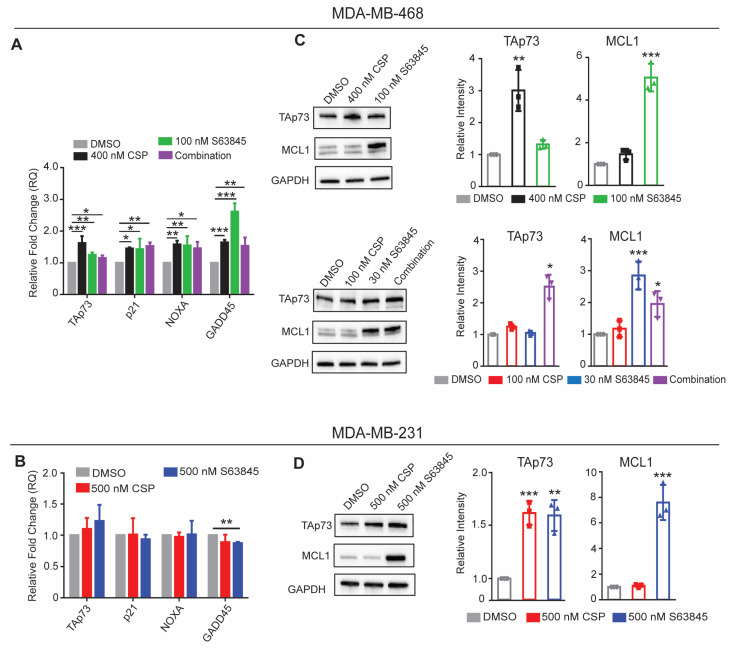
Cisplatin and S63845 induce TAp73 expression and downstream targets in MDA-MB-468 cells. (**A**,**B**) RT-qPCR analysis of MDA-MB-468 and MDA-MB-231 cells treated over various conditions of cisplatin and S63845 for 24 h, respectively. Western blot analyses of (**C**) MDA-MB-468 and (**D**) MDA-MB-231 cells treated over various conditions of cisplatin and S63845 for 24 h. Combination treatment consisting of 100 nM cisplatin and 30 nM S63845. Differences between groups were evaluated using a matched-pair two-way ANOVA followed by Tukey’s post hoc test; *p* < 0.05. * *p* < 0.05; ** *p* < 0.01; *** *p* < 0.001. All experiments were performed in biological triplicate.

**Figure 7 cancers-15-04481-f007:**
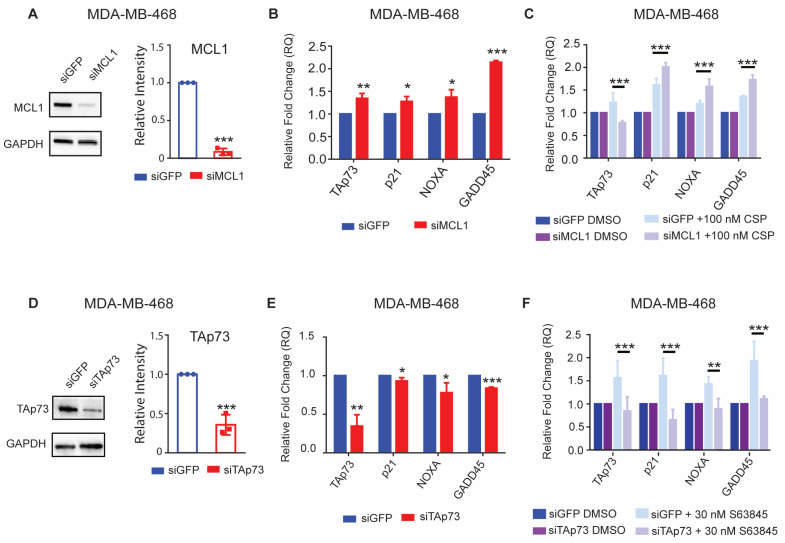
MCL1 and TAp73 mediate synergistic gene expression of TAp73 and downstream targets. (**A**) Representative Western blot of MDA-MB-468 cells treated with siGFP or siMCL1. RT-qPCR analysis of (**B**) MDA-MB-468 cells transfected with siGFP or MCL1 for 24 h. RT-qPCR analysis of (**C**) MDA-MB-468 cells transfected with siGFP or MCL1 for 48 h and treated with 100 nM cisplatin for 24 h. (**D**) Representative Western blot of MDA-MB-468 cells treated with siGFP or siTAp73. RT-qPCR analysis of (**E**) MDA-MB-468 cells transfected with siGFP or TAp73 for 24 h. RT-qPCR analysis of (**F**) MDA-MB-468 cells transfected with siGFP or TAp73 for 48 h and treated with 30 nM S63845 for 24 h. Gene expression was normalized to the respective siRNA target treated with DMSO. Differences between groups were evaluated using a matched-pair two-way ANOVA followed by Tukey’s post hoc test; *p* < 0.05. * *p* < 0.05; ** *p* < 0.01; *** *p* < 0.001. All experiments were performed in biological triplicate.

## Data Availability

Any data not directly reported in the manuscript may be requested from the corresponding authors.

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
