# Peer review of "Myeloid Cell Leukemia 1 Small Molecule Inhibitor S63845 Synergizes with Cisplatin in Triple-Negative Breast Cancer"

_cancers, 2023, doi:10.3390/cancers15184481_

Round 1
Reviewer 1 Report
In this manuscript, the authors demonstrated that S63845 synergizes with cisplatin in triple negative breast cancer. Overall, the findings are interesting. However, several issues should be addressed before the manuscript can be considered for publication.
Major points:
1. Why did the expression of the target genes increase while the expression of TAp73 decreased when both siMCL1 and 100 nM CSP were co-administered in Figure 7?
2. The authors should present plausible reasons for the observed synergistic effect when combining S63845 and cisplatin, which was only observed in MDA-MB-468 and HCC1143, in the "Discussion" section.
3. The authors should conduct a similar analysis in Figure 7 using MDA-MB-231 cells to investigate the resistance to cisplatin and S63845 in those cells.
Minor points:
1. Due to the mixed usage of the notations TP73 and TAp73, which leads to confusion, it is necessary to provide an elucidation on the difference between TP73 and TAp73 in the "Introduction" section.
2. In page 4 line 140, "B-mercaptoethanol" should be "β-mercaptoethanol".
3. In page 4 line 144, please provide an explanation of PBST.
4. In page 6 line 241, "uM" should be "μM".
5. Due to inaccuracies in the content of the legends for Figures 4 and 6, please make the necessary corrections.
6. In page 12 line 402, please provide citations to the previous studies.
Author Response
Reviewer #1
Reviewer Comment- Why did the expression of the target genes increase while the expression of TAp73 decreased when both siMCL1 and 100 nM CSP were co-administered in Figure 7?
Response- Target gene increase was expected as treatment with either CSP or siMCL1 are expected to increase TAp73 activity either by activation or loss of suppression, respectively. The decrease in TAp73 mRNA was not anticipated and may be due to feedback regulation resulting from increased TAp73 activity. A number of prior studies have found that TAp73 target genes can regulate TAp73 expression, though this is variable and is cell line-specific.
Reviewer Comment- The authors should present plausible reasons for the observed synergistic effect when combining S63845 and cisplatin, which was only observed in MDA-MB-468 and HCC1143, in the "Discussion" section.
Response- We believe that the synergy is due to the recently described anti-p73 function of MCL1 as outlined in the introduction (lines: 92-100). This observation is what served as the foundation for the present study. Further, we suggest in the discussion that the synergy is observed in these two cell lines due to their classification as basal-like TNBCs (lines: 489-501). Other cell lines that did not exhibit synergy are classified as mesenchymal and are not sensitive to single agent MCL1 inhibition.
Reviewer Comment- The authors should conduct a similar analysis in Figure 7 using MDA-MB-231 cells to investigate the resistance to cisplatin and S63845 in those cells.
Response- We agree that further analysis of resistance mechanisms mediated by MCL1 could be of interest in mesenchymal TNBC, but chose to move forward with analysis of synergy only in lines responsive to single agent in this initial study.
Reviewer Comment- Due to the mixed usage of the notations TP73 and TAp73, which leads to confusion, it is necessary to provide an elucidation on the difference between TP73 and TAp73 in the "Introduction" section.
Response- We apologize for the oversight and have added the necessary introduction of TAp73 as an isoform of TP73 (lines: 81-84).
Reviewer Comment- In page 4 line 140, "B-mercaptoethanol" should be "β-mercaptoethanol".
Response- B-mercaptoethanol was changed to "-mercaptoethanol on line 144 in manuscript.
Reviewer Comment- In page 4 line 144, please provide an explanation of PBST.
Response- PBST was changed to phosphate buffered saline + tween (PBST) in manuscript on line 149
Reviewer Comment- In page 6 line 241, "uM" should be "μM".
Response- We apologize for this mistake in the original submission, we have changed uM to μM (lines 247-248)
Reviewer Comment- Due to inaccuracies in the content of the legends for Figures 4 and 6, please make the necessary corrections.
Response- We apologize for this mistake in the original submission. Both legends for Figure 4 and 6 have been updated in the text to accurately represent each graph and experimental technique.
Reviewer Comment- In page 12 line 402, please provide citations to the previous studies.
Response- We apologize that we missed this reference in the original submission, two references have been added to support authors claim based off prior studies (line 425; references 24 and 34).
Reviewer 2 Report
The subject addressed in this article is interesting because there is a huge unmet need to uncover anticancer drug resistance therapeutics. I found this MS interesting and well presented. I suggest acceptance with minor revision keeping in consideration the comments given below:
Suggestions:
· Please support MTS results with colony formation studies in figure1.
· Protein levels need to supports claims like cell lines exhibit 235 amplifications of MCL1”.
· Though authors have provided gene silencing and qPCR results but to reach to any conclusion, authors need to analyze the mechanism of action with genetic manipulation both TP73 and MCL1.
· As compare to bar graph, line plot would be easier to understand for figure 3E.
· In figure 4F, CCC3 protein bands are not uniform with other CCC3 blot results so authors need to work on that and it would be better if authors can provide both pro and cleaved casp3 blot.
Author Response
Reviewer #2
Reviewer Comment- Please support MTS results with colony formation studies in figure 1.
Response- The colony formation assays presented in Figure 5 were performed to support the synergistic compound screenings guided from the initial MTS screenings.
Reviewer Comment- Protein levels need to supports claims like cell lines exhibit 235 amplifications of MCL1”.
Response- We apologize for the omission of the reference where this observation was initially made. This has been included (line: 242, reference 20).
Reviewer Comment- Though authors have provided gene silencing and qPCR results but to reach to any conclusion, authors need to analyze the mechanism of action with genetic manipulation both TP73 and MCL1.
Response- Both p73 and MCL1 have external roles outside of apoptosis and cell cycle arrest. Further, both p73 and MCL1 are essential genes in human development and stem cell differentiation, and long-term genetic manipulation can postulate an array of altered gene expression that could confound our results. This is the reason we chose to transiently downregulate these genes using siRNA.
Reviewer Comment- As compared to bar graph, line plot would be easier to understand for figure 3E.
Response- Our data in Figure 3E represents various treatment conditions and cell cycle stages, therefore we believe a bar graph better represents the differences in the total cell population. We are including a variant of Figure 3E for review only, but feel the bar plot is more accessible for a broad audience.
Reviewer Comment- In figure 4F, CCC3 protein bands are not uniform with other CCC3 blot results so authors need to work on that, and it would be better if authors can provide both pro and cleaved casp3 blot.
Response- We concur that the difference cell lines have difference CCC3 banding patterns and are not sure how to modify this as this is a known variance that can be observed between different cell lines. We regret that we did not collect the pro-casp3 blots when this study was initially conducted.
Reviewer 3 Report
The treatment of Triple negative breast cancer (TNBC) remains an important clinical problem. In this paper, Acton and Plazek report that the MCL1 small molecule inhibitor S63845 synergizes with cisplatin to arrest cell growth and induce apoptosis. Previous work has shown that MCL1 expression induces resistance to platinum-based chemotherapy. The authors combined S63852 with cisplatin and determined that the combination induces apoptosis, while also decreasing cell proliferation in a subset of TNBC cell lines. This effect is mediated with TAp73. The authors suggest that this combination may be useful for treatment of TNBC. In general, this is a clearly written and logically presented study that should be of interest to your readers. However, revision is needed in several areas of data presentation and experimental design to strengthen the manuscript. Specific points include the following:
Materials and Methods
In general, the authors do not need to provide many of the extensive details for this study-for example line 106 cells were rinsed with 5 ml of DPBS.
Line 112 Please state why MDA-MB-468 and MDA-MB-231 cell lines were used.
It would be useful to classify the cell lines as mesenchymal or basal-like subtypes at the beginning of the study.
Colony formation Assay: Please provide details on how the colonies were quantitated.
Line 135 The authors state that a 10-point dose response curve were used, but the Results show a 9- point study.
Results
Figure 1 Line 232 Please indicate which cell lines are TP53 wild type or mutated.
Figure 3 Please provide the reference for line 283.
Fig. 5 Have the colony assays been carried out with a variety of drug combinations? If so, are the same synergistic effects observed?
The Legend for Fig. 6 needs to be corrected. For example, 6b is described as Western Blot analysis of MDA-MB-468 cells.
Do the authors have Western blots for p21, NOXA and GADD45 for this set of Figures?
Figure 6 Line 394 The authors state the MDA-MB- 468 cells were treated over varying concentrations of cisplatin, but only one concentration is shown in 6A.
Do the authors have Western blots for p21, NOXA and GADD45 for Figure 6A?
The Western blot with 400 nM CSP and 100 NM of S63845 needs a lane for the combination.
Fig. 7C It looks as if there is no increase in TAP73 after siMCL1 treatment in the right-hand panel of Fig. 7B as opposed to the left-hand panel. There is the same concern with 7F.
Minor comments
Introduction Line 78 The authors describe the two studies (31 and 32) as recent, although they were published in 2004 and 2000.
Line 90 Please provide the references for the studies cited in this sentence.
Line 152 Do the authors mean antibodies instead of dilution?
Author Response
Reviewer #3
Reviewer Comment- In general, the authors do not need to provide many of the extensive details for this study-for example line 106 cells were rinsed with 5 ml of DPBS.
Response- While we understand that extensive methods may be unnecessary, we provided these details in hopes of reproducibility by other research groups if a follow up study is performed.
Reviewer Comment- Line 112 Please state why MDA-MB-468 and MDA-MB-231 cell lines were used.
Response- We have added a statement outlining the criteria used to select these cell lines for interrogation in section 3.1 (lines: 237-248). In addition we highlight that MDA-MB-468 and MDA-MB-231 are chosen for their sensitivity or resistance, respectively (line 114), to enable us to analyze the impact on compound viability.
Reviewer Comment- It would be useful to classify the cell lines as mesenchymal or basal-like subtypes at the beginning of the study.
Response- The manuscript Results 3.1 section has been updated to include the basal and mesenchymal classifications of each of the cell lines used in this study.
Reviewer Comment- Colony formation Assay: Please provide details on how the colonies were quantitated.
Response- The method section has been updated to include that images were quantified on Fiji Image J and colonies were considered for analysis with >5 cells (lines: 133-134).
Reviewer Comment- Line 135 The authors state that a 10-point dose response curve were used, but the Results show a 9- point study.
Response- We apologize for this mistake, “10-point dose response curve” has been replaced with “9-point dose response curve” (line 138).
Reviewer Comment- Figure 1 Line 232 Please indicate which cell lines are TP53 wild type or mutated.
Response- p53 status (mutation or wildtype) of studied cell lines have been added in section 3.1 (lines 240-241)
Reviewer Comment- Figure 3 Please provide the reference for line 283.
Response- Our prior study reference has been added in the text of the manuscript (line 293, reference 33).
Reviewer Comment- Fig. 5 Have the colony assays been carried out with a variety of drug combinations? If so, are the same synergistic effects observed?
Response- The treatment groups in this assay are the same combination studies that are implemented in other assays. There is a significant difference in total formed colonies between combination and single agent treated groups. We did not conduct a full isobologram analysis using colony formation.
Reviewer Comment- The Legend for Fig. 6 needs to be corrected. For example, 6b is described as Western Blot analysis of MDA-MB-468 cells.
Response- We apologize for this mistake in the original submission. The legend for Figure 6 has been updated in the text to accurately represent each graph and experimental technique.
Reviewer Comment- Do the authors have Western blots for p21, NOXA and GADD45 for this set of Figures?
Response- While we are aware that mRNA and protein expression are not always linear, this study was focused on MCL1’s impact on p73’s transcriptional activity. We did not collect WB data for these targets with these samples.
Reviewer Comment- Figure 6 Line 394 The authors state the MDA-MB- 468 cells were treated over varying concentrations of cisplatin, but only one concentration is shown in 6A.
Response- We apologize for this confusion within the text. The single agents of cisplatin and S63845 are 400 nM and 100 nM, respectively. However, the combination group is comprised of 100 nM cisplatin and 30 nM S63845, in line with other assays.
Reviewer Comment- Do the authors have Western blots for p21, NOXA and GADD45 for Figure 6A?
Response- Response- While we are aware that mRNA and protein expression are not always linear, this study was focused on MCL1’s impact on p73’s transcriptional activity. We did not collect WB data for these targets with these samples.
Reviewer Comment- The Western blot with 400 nM CSP and 100 NM of S63845 needs a lane for the combination.
Response- This study was investigating the combination of the lower doses of cisplatin and S63845 in combination as opposed to the higher doses. Combination of the higher concentrations of cisplatin and S63845 causes significant death, and we were unable to obtain lysate for these experiments at 24 hours.
Reviewer Comment- Fig. 7C It looks as if there is no increase in TAP73 after siMCL1 treatment in the right-hand panel of Fig. 7B as opposed to the left-hand panel. There is the same concern with 7F.
Response- For the first portion of this, we have responded to a similar question by Reviewer 1 (1st comment). For the second half, 7F shows treatment with siTAp73 and thus we would expect to see a decrease in TAp73 gene expression.
Reviewer Comment- Introduction Line 78 The authors describe the two studies (31 and 32) as recent, although they were published in 2004 and 2000.
Response- We have changed the sentence introduction on line 78 from “Recent” to “Prior”
Reviewer Comment- Line 90 Please provide the references for the studies cited in this sentence.
Response- We apologize for this missed citation; the reference has been added to the manuscript text (line 93, reference 33).
Reviewer Comment- Line 152 Do the authors mean antibodies instead of dilution?
Response- Line 157 has been changed to “antibody dilutions” as opposed to just “dilution”.
Round 2
Reviewer 3 Report
The authors have done a good job in replying to the previous critiques. The manuscript should now be acceptable for publication.